

# Sarcopenia as an important determinant for adverse outcomes in patients with pyogenic liver abscess

Li Liu[1,2,*], Shaohua Liu[1,2,*], Meng Hao[3], Song Hu[4], Tian Yu[1,2], Yunkai Yang[5] and Zhelong Liu[1,2]

[1] Department of Endocrinology, Tongji Hospital, Tongji Medical College, Huazhong University of Science and Technology, Wuhan, China
[2] Branch of National Clinical Research Center for Metabolic Diseases, Wuhan, Hubei, China
[3] Department of Gastroenterology, Zigui County People's Hospital, Yichang, China
[4] Department of Gastroenterology, Tongji Hospital, Tongji Medical College, Huazhong University of Science and Technology, Wuhan, China
[5] Eight-year Program of Clinical Medicine, Tongji Hospital, Tongji Medical College, Huazhong University of Science and Technology, Wuhan, China
* These authors contributed equally to this work.

## ABSTRACT

**Background**. Low muscle mass/sarcopenia has been associated with poor prognosis in many diseases, but its clinical significance in pyogenic liver abscess (PLA) remains unclear. The purpose of this study is to investigate the relationship between muscle mass and prognosis of patients with PLA.

**Methods**. A total of 154 adult patients with PLA hospitalized at Tongji Hospital (Wuhan, Hubei, China) between October 2011 and June 2021 were included in this retrospective analysis. Muscle-fat related indicators were measured by computed tomography (CT) images at the third lumbar vertebra (L3) level. The data of patients between the sarcopenia group and non-sarcopenia group were compared. Multivariate logistic regression and receiver operating characteristic (ROC) curve analyses were performed.

**Results**. The skeletal muscle index (SMI) was independently associated with adverse outcomes (95% CI [0.649–0.954], $P = 0.015$) of PLA in multivariate logistic regression analysis. This conclusion held true in sex-specific subgroup analysis. ROC analysis indicated that SMI may predict adverse outcomes in both male (area under the ROC curve [AUC], 0.718; cut-off, 52.59; $P < 0.001$) and female (AUC, 0.714; cut-off, 38.39; $P = 0.017$) patient populations.

**Conclusions**. Sarcopenia serves as an independent risk factor for poor prognosis in PLA and patients with sarcopenia may be more prone to adverse outcomes.

# INTRODUCTION

Pyogenic liver abscess (PLA) is the most common visceral abscess clinically. It can be defined as suppurative infection of the liver parenchyma, which may originate from the

Corresponding author
Zhelong Liu, liuzhelong@163.com

biliary tract, portal vein, or hematogenic or cryptogenic, and adjacent structure infections (*Song et al., 2020*). The incidence of PLA is higher in Asian countries, with reported rates of 12–18 cases per 100,000 people per year, with a mortality of approximately 2%–31% (*Lo et al., 2015*; *Poovorawan et al., 2016*). The onset of the disease is sudden and complex, often leading to liver necrosis, septic shock and other serious consequences, posing a serious threat to individual life. Therefore, identifying prognostic risk factors is crucial to providing patients with timely and effective interventions to improve their poor prognosis, which is of great clinical significance.

Body composition, including muscle and fat mass, has been increasingly linked to the clinical course, treatment response, and prognosis of various diseases (*Grillot et al., 2020*; *Hung et al., 2021*; *Kang et al., 2019*; *Kang et al., 2018*). Sarcopenia, which is characterized by a loss of skeletal muscle mass and strength due to various reasons, was first proposed by *Rosenberg (1989)* and *Chen et al. (2014)*. The presence of low muscle mass is often accompanied by metabolic, physical and functional disabilities, which can impact the prognosis of various diseases (*Cruz-Jentoft et al., 2010*; *Janssen, Heymsfield & Ross, 2002*). A retrospective study of 88 Crohn's disease patients by *Grillot et al. (2020)* revealed that sarcopenia was linked to adverse outcomes in severe Crohn's disease cases. In addition, sarcopenia is thought to be closely related to infection because it may lead to an immunosuppressive state in which the body's ability to fight against infection is weakened (*Kaido et al., 2013*; *Schaible & Kaufmann, 2007*). The results of a large-scale multi-center study on patients with splenic abscess suggest a strong association between sarcopenia and poor prognosis, specifically hospital mortality, in such patients (*Hung et al., 2021*).

Computed tomography (CT) is capable of distinguishing different types of body tissues (*e.g.*, fat, muscle, bone) based on tissue-specific attenuation values, making it possible to quantify skeletal muscle. CT has been recognized by the European Working Group on Sarcopenia in Older People (EWGSOP) as the gold standard for measuring muscle content, and is often used for the diagnosis of sarcopenia (*Chen et al., 2014*; *Cruz-Jentoft et al., 2019*). In particular, the CT images of specific lumbar markers, particularly at the L3 level, have shown a significant association with whole-body muscle mass. This imaging method has been widely utilized to assess low muscle mass in various diseases, proving its effectiveness in predicting prognosis even in patients with normal or high weight (*Cruz-Jentoft et al., 2019*). Several risk factors have been identified for the prognosis of PLA, including malnutrition (*Xu, Zhou & Zheng, 2019*), pleural effusion the size of abscess, and microbiology (*Chan, Chia & Shelat, 2022*; *Lee et al., 2021*; *Xu, Zhou & Zheng, 2019*). Few studies have examined the association between sarcopenia and adverse outcomes in PLA patients. Thus, the aim of the present study was to investigate the association between the sarcopenia, which was determined by measuring the body composition of a single layer CT slice at the L3 level, and adverse outcomes of PLA patients.

## MATERIALS & METHODS

### Study design and participants

Clinical medical data of a consecutive case series of 458 patients with PLA admitted to Tongji Hospital, Tongji Medical College, Huazhong University of Science and Technology, between October 2011 and June 2021, were retrospectively collected from medical record system (search term "pyogenic liver abscess", ICD 10 code = K750). The inclusion criteria for PLA for this study were as follows: (1) the diagnosis of PLA required the presence of at least one lesion observed through liver imaging (magnetic resonance imaging, CT, or ultrasound), along with either evidence of lesion resolution following antimicrobial therapy or a positive blood/pus culture; (2) age ≥18 years; (3) availability of complete key laboratory results and images of abdominal CT scans; (4) absence of amebic liver abscess or parasitic liver abscess. Finally, a total of 154 patients who underwent abdominal CT scans that could be used to assess sarcopenia and had ruled out underlying conditions that put them at risk of muscle loss (stroke sequelae, spinal cord diseases, peripheral neuropathy and other neurological diseases lead to decreased muscle strength, wasting diseases like malignant tumors and tuberculosis) were included in this analysis.

The Ethics Committee of Tongji Hospital, Huazhong University of Science and Technology, approved this study, which was exempt from informed consent because of its retrospective design (TJ-IRB20230118).

### Demographic and clinical variables

Data collected from the hospital's electronic medical record system included information on patients' demographic characteristics, comorbidities, clinical symptoms and signs, vital signs upon admission, laboratory tests, the microbial culture results of blood or pus, imaging findings, treatment, and adverse outcomes including mortality and serious complications. By incorporating previous literature (*Xu & Wang, 2019*) and considering the specific circumstances of our study, we defined serious complications in our study as septic shock (defined as acute circulatory failure with uncorrectable hypotension unexplained by other causes, despite sufficient fluid resuscitation (*Singer et al., 2016*)), acute renal injury (indicated by serum creatinine above 176 μ mol/L, or an absolute increase was greater than 44 μ mol/L (*Englberger et al., 2011*)), acute hepatic injury (defined according to the WHO diagnostic criteria, it is characterized by elevated levels of alanine aminotransferase (ALT), aspartate aminotransferase (AST), or total bilirubin (TBIL), where any one of these markers exceeds 1.25 times the upper limit of the reference value), heart failure (defined in accordance with the guidelines of the Heart Failure Association of the European Society of Cardiology (*Mueller et al., 2019*)), myocardial infarction (defined as a serum level of high-sensitivity cardiac troponin I (hs-cTnI) > 34 pg/mL (*Gao et al., 2020*)), pulmonary edema, lung infection (diagnosed on the basis of pulmonary imaging findings), and acute respiratory distress syndrome (ARDS) (defined according to The Berlin Definition of Acute Respiratory Distress Syndrome (*Ranieri et al., 2012*)).

## Selection of CT images

Non-contrast CT scans with a 0.5 cm slice thickness were used from the abdominal CT scans with 64 row detector configuration, 0.5 s/rotation, and 120 kV tube voltage. All CT images were acquired from the Image Archiving and Communication System, anonymized and viewed using Sante DICOM Viewer software. Due to the close relationship between muscle and fat, and the fact that obesity and sarcopenia often co-exist and aggravate each other, leading to a variety of diseases (*Hong & Choi, 2020*), we also measured fat-related indicators in addition to muscle. Moreover, it has been reported that adipose-related indicators are independent risk factors for prognosis of many diseases (*Gonçalves, Magro & Martel, 2015*; *Goto et al., 2021*; *Valencia et al., 2020*; *Yu et al., 2022*). The cross-sectional area of skeletal muscle and adipose tissue on single-slice CT scans at the L3 level is currently accepted as the best representatives of the whole body, as determined by professional radiologists. L3 segments were identified progressively starting with the first thoracic vertebra, and if the first thoracic vertebra was not included, the 12th thoracic vertebra and the sacral joint were used to assist in positioning. If abdominal CT examinations were repeated during hospitalization, only the first examination after admission was considered for analysis.

## Image analysis

An experienced radiologist conducted the CT image measurements. The radiologist performed random measurements of all the patients' images without access to any clinical data or final diagnosis. To ensure the reliability and stability of our data, the radiologist conducted a second round of random measurements on all CT images after a 2-week interval. Afterwards, we performed a consistency test on the data from the two measurements and observed a very high level of agreement (all intraclass correlation coefficients were greater than or equal to 0.996, $P < 0.001$). As a result, we decided to use the data from the first measurement for our analysis. The following parameters were measured using a semi-automatic software named slice-Omatic V5.0 (TomoVision, Magog, Canada): skeletal muscle area (SMA, cm$^2$), skeletal muscle density (SMD, HU), and fat-related indicators, such as intramuscular adipose tissue (IMAT, cm$^2$), subcutaneous fat area (SFA, cm$^2$), and visceral fat area (VFA, cm$^2$). The SMA at the L3 level was determined by measuring the combined areas of the psoas major, quadratus lumborum, paraspinal muscles, transversus abdominis, rectus abdominis, and internal and external oblique muscles. The areas measured were then adjusted for height square (m$^2$) to obtain the skeletal muscle index (SMI, cm$^2$/m$^2$), subcutaneous adipose index (SAI, cm$^2$/m$^2$), and visceral adipose index (VAI, cm$^2$/m$^2$).The region of interest (ROI) was determined by applying predefined radiation attenuation thresholds, as illustrated in Fig. 1A. The attenuation thresholds of skeletal muscle were −29 to 150 HU, for intramuscular adipose tissue and for subcutaneous adipose tissue were −190 to −30 HU, and visceral fat tissue were −150 to −50 HU. SMD will be displayed automatically after the ROI of skeletal muscle is selected (*Steele et al., 2021*). As previously reported, sarcopenia was defined as SMI < 52.4 cm$^2$/m$^2$ for males and <38.5 cm$^2$/m$^2$ for females (*Amini et al., 2019*; *Grillot et al., 2020*; *Kang et al., 2018*). The mesenteric fat index (MFI) was calculated as the ratio of VFA to SFA (*Erhayiem et al., 2011*). CT images at the L3 level were also used to measure abdominal wall and

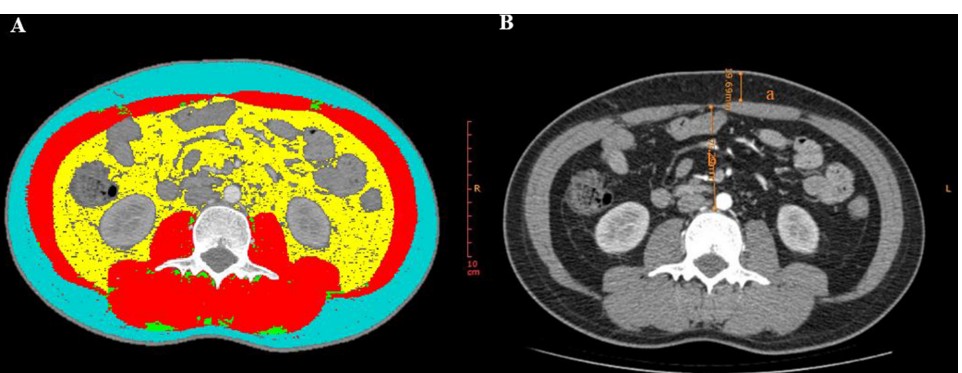

**Figure 1** **CT parameters at the L3 level measured by Slice-Omatic software.** (A) Measurement of muscle and fat area. Red: skeletal muscle area (SMA); Blue: subcutaneous fat area (SFA); Green: intramuscular adipose tissue (IMAT); Yellow: visceral fat area (VFA). (B) a: abdominal wall fat thickness, b: intraabdominal fat thickness.

intra-abdominal fat thickness, with abdominal wall thickness determined by measuring the anteroposterior distance between the skin and the anterior rectus sheath (Fig. 1B-a) and the intra-abdominal fat thickness defined as the distance between the linea alba and the posterior aortic wall (Fig. 1B-b) (*Jin et al., 2021*).

## Statistical analysis

The modified Kolmogorov–Smirnov test was utilized to assess the normal distribution of continuous variables. Continuous variables with a normal distribution presented as mean ± standard error, while non-normal distribution variables presented as median (quartile distance). Categorical variables presented as counts (percentages). Student's $t$-test or Mann–Whitney U test was employed to compare continuous variables between groups (sarcopenia group *vs.* non-sarcopenia group), while Fisher's exact test or Chi-square test was used for categorical variables. The distribution of isolated microorganisms was depicted using pie chart. Logistic regression analysis was conducted to identify risk factors that might be associated with adverse outcomes (serious complications and mortality) of PLA. Variables without collinearity (correlation coefficient < 0.5) were selected for the multivariate analysis, considering their clinical implications and statistical significance ($P < 0.05$) obtained from the univariate logistic regression analysis. Receiver operating characteristic curve (ROC) analysis was performed to evaluate the predictive performance for adverse outcomes based on the area under the ROC curve (AUC). Statistical analysis and graph construction were conducted using SPSS Version 20.0 software (SPSS Inc., Chicago, Illinois, USA) and GraphPad Prism (ver.9, GraphPad Software, La Jolla, CA, USA). A two-sided $P < 0.05$ was considered statistically significant.

## RESULTS

### Demographics and baseline characteristics of patients with PLA

The demographic data of patients at admission was shown in Table 1. A total of 154 patients with PLA were enrolled in this study, with an average age of 53.8 ± 1.1 years. Of these

patients, 111 (72.1%) were male. The most common clinical symptom was abdominal distension or abdominal pain ($n = 71$, 46.1%), followed by fatigue and muscle pain ($n = 64$, 41.6%) and fever ($n = 46$, 29.9%). The most common comorbidity was liver and gallbladder stones ($n = 33$, 21.4%), followed by hypertension ($n = 28$, 18.2%) and diabetes ($n = 25$, 16.2%). All patients with PLA were classified into the sarcopenia group and the non-sarcopenia group based on the diagnostic criteria for sarcopenia described previously. Males were predominant in the sarcopenia group and the trend was more pronounced than in the non-sarcopenia group. The clinical symptoms/signs, co-morbidity, and vital sign at admission did not differ significantly between the sarcopenia and non-sarcopenia groups with PLA. In terms of liver abscess features, *Klebsiella pneumoniae* was more likely to be present in the sarcopenia group among patients with positive microbial cultures. However, abscess size and gas formation did not show significant differences between the sarcopenia and non-sarcopenia groups.

### Baseline laboratory parameters and treatments of patients with PLA

The results presented in Table 2 indicate that the sarcopenia group exhibited significantly higher levels of inflammatory markers, including C-reactive protein (CRP, $P = 0.030$) and procalcitonin (PCT, $P = 0.011$), compared to the non-sarcopenia group. Moreover, the sarcopenia group showed significantly increased levels of various hepatic function indicators, such as TBIL ($P = 0.048$), direct bilirubin (DBIL, $P = 0.002$), ALT ($P = 0.017$), AST ($P = 0.017$), alkaline phosphatase (ALP, $P = 0.020$), and gamma glutamyl transpeptidase ($\gamma$-GT, $P = 0.032$) compared with the non-sarcopenia group in patients with PLA.

Out of the total 154 patients with PLA, 71 (46.1%) showed positive microbial culture results. As shown in Fig. S1, the sarcopenia group cultivated a greater variety of microorganisms than the non-sarcopenia group. *Klebsiella pneumoniae* was the most common pathogenic microorganisms in both the sarcopenia and non-sarcopenia groups, accounting for 46.81% and 79.17%, respectively, followed by *Escherichia coli* (17.02% and 4.17%), *Streptococcus* (10.64% and 8.97%), and *Staphylococcus* (10.64% and 4.14%).

Regarding the treatment (Table 2), 42 patients (27.3%) received conservative treatment with antibiotics alone, 96 patients (62.3%) received antibiotics with abscess puncture and drainage, and 9 patients (5.8%) received antibiotics with surgery. Glucocorticoid use was observed more frequently in the sarcopenia group ($P = 0.045$). There were no significant differences between the sarcopenia and non-sarcopenia groups regarding other treatments received.

### CT parameters based on muscular and adipose indicators of patients with PLA

The results presented in Table 3 show that in both male and female groups, the sarcopenia groups had lower SMA ($P < 0.001$) and SMI ($P < 0.001$) compared to non-sarcopenia groups. Interestingly, the male sarcopenia group had significantly lower indices related to fat, including SFA ($P = 0.004$), VAI ($P = 0.027$), SAI ($P = 0.003$), and abdominal wall fat thickness ($P = 0.039$) compared to the non-sarcopenia group. However, no such differences were observed in the female group.

**Table 1  Demographics and baseline characteristics between sarcopenia and non-sarcopenia groups in patients with pyogenic liver abscess.**

| Characteristics | Total (n = 154) | Sarcopenia (n = 91) | Non-sarcopenia (n = 63) | P value |
|---|---|---|---|---|
| Age (years) | 53.8 ± 1.1 | 54.7 ± 1.5 | 52.5 ± 1.4 | 0.324 |
| Male, n (%) | 111 (72.1) | 72 (79.1) | 39 (61.9) | 0.020 |
| BMI | 23.4 (21.2, 25.7) | 22.4 (19.4, 24.6) | 24.9 (22.8, 28.0) | <0.001 |
| Clinical symptoms/signs | | | | |
| Fever, n (%) | 46 (29.9) | 31 (34.1) | 15 (23.8) | 0.211 |
| Nausea and vomiting, n (%) | 28 (18.2) | 19 (20.9) | 9 (14.3) | 0.396 |
| Abdominal distension or pain, n (%) | 71 (46.1) | 43 (47.3) | 28 (44.4) | 0.745 |
| Diarrhea, n (%) | 13 (8.4) | 6 (6.6) | 7 (11.1) | 0.383 |
| Fatigue and muscle pain, n (%) | 64 (41.6) | 38 (41.8) | 26 (41.3) | 0.952 |
| Chest pain, n (%) | 8 (5.2) | 6 (6.6) | 2 (3.2) | 0.347 |
| Palpitation, n (%) | 13 (8.4) | 8 (8.8) | 5 (7.9) | 0.851 |
| Cough and sputum, n (%) | 15 (9.7) | 7 (7.7) | 8 (12.7) | 0.303 |
| Dizziness or headache, n (%) | 20 (13.0) | 11 (12.1) | 9 (14.3) | 0.690 |
| Dyspnea, n (%) | 3 (1.9) | 3 (3.3) | 0 (0.0) | 0.146 |
| Co-morbidity | | | | |
| Diabetes, n (%) | 25 (16.2) | 15 (16.5) | 10 (15.9) | 0.920 |
| Hypertension, n (%) | 28 (18.2) | 19 (20.9) | 9 (14.3) | 0.297 |
| Cardiovascular disease, n (%) | 6 (3.9) | 5 (5.5) | 1 (1.6) | 0.218 |
| Chronic respiratory disease, n (%) | 7 (4.5) | 2 (2.2) | 5 (7.9) | 0.093 |
| Liver and gallbladder stones, n (%) | 33 (21.4) | 21 (23.1) | 12 (19.0) | 0.549 |
| Viral hepatitis, n (%) | 24 (15.6) | 15 (16.5) | 9 (14.3) | 0.712 |
| Fatty liver disease, n (%) | 17 (11.0) | 8 (8.8) | 9 (14.8) | 0.285 |
| Vital sign at admission | | | | |
| Temperature, °C | 36.7 (36.4, 37.5) | 36.6 (36.4, 37.7) | 36.8 (36.5, 37.1) | 0.529 |
| Systolic blood pressure, mmHg | 121 ± 1.6 | 119 ± 2.2 | 123 ± 2.5 | 0.269 |
| Diastole blood pressure, mmHg | 76 ± 1.0 | 74 ± 1.3 | 78 ± 1.5 | 0.064 |
| Heart rates, /min | 88 (78, 102) | 86 (78, 102) | 88 (80, 98) | 0.975 |
| Respiratory rates, /min | 20 (20, 20) | 20 (20, 20) | 20 (20, 20) | 0.320 |
| The size of abscess (≥5 cm) | 73 (47.4) | 40 (44.0) | 33 (52.4) | 0.587 |
| Gas formation | 18 (11.7) | 13 (14.3) | 5 (7.9) | 0.228 |
| Klebsiella pneumonia[*] | 41 (57.8) | 22 (46.8) | 19 (79.2) | 0.009 |

**Notes.**

Data are presented as mean ±SE or median (interquartile range) for continuous variables and n (%) for categorical variables.

*P*-values comparing sarcopenia group and non-sarcopenia group are from Student's *t*-test, Mann–Whitney U-test, $\chi 2$ test, or Fisher's exact test.

[*]There were a total of 71 patients with microbial culture results, including 47 in the sarcopenia group and 24 in the non-sarcopenia group.

Abbreviations: BMI, body mass index.

**Table 2 Laboratory indices and treatments between sarcopenia and non-sarcopenia groups in patients with pyogenic liver abscess.**

| Variables | Normal range | Total ($n = 154$) | Sarcopenia ($n = 91$) | Non-sarcopenia ($n = 63$) | P value |
|---|---|---|---|---|---|
| White blood cell, $10^9$/L | 3.5–9.5 | 10.9 (7.8, 15.0) | 11.1 (7.4, 14.9) | 10.8 (8.0, 15.6) | 0.569 |
| Neutrophil count, $10^9$/L | 1.8–6.3 | 9.1 (5.8, 13.0) | 9.0 (5.6, 12.7) | 9.1 (5.9, 13.8) | 0.558 |
| Lymphocyte count, $10^9$/L | 1.1–3.2 | 1.1 (0.7, 1.5) | 1.1 (0.7, 1.4) | 1.2 (0.8, 1.7) | 0.162 |
| Hemoglobin, g/L | 115–150 | 114 ± 1.8 | 113 ± 2.3 | 116 ± 2.9 | 0.477 |
| Platelet count, $10^9$/L | 125–350 | 240 ± 11.4 | 234 ± 15.8 | 249 ± 15.8 | 0.517 |
| CRP, mg/L | <1 | 134.1 ± 9.7 | 152.7 ± 13.7 | 111.4 ± 12.8 | 0.030 |
| PCT, ng/ml | <0.05 | 0.93 (0.17, 6.59) | 1.38 (0.34, 9.85) | 0.39 (0.07, 2.43) | 0.011 |
| Total bilirubin, μmol/L | <=21 | 13.0 (8.6, 19.4) | 14.4 (9.3, 23.4) | 11.8 (8.0, 16.4) | 0.048 |
| Direct bilirubin, μmol/L | <=8 | 5.7 (3.2, 10.5) | 6.3 (4.8, 13.7) | 4.8 (2.9, 6.9) | 0.002 |
| ALT, U/L | <33 | 37 (21, 71) | 46 (22, 90) | 28 (16, 53) | 0.017 |
| AST, U/L | <32 | 29 (19, 60) | 37 (22, 68) | 23 (16, 42) | 0.003 |
| ALP,U/L | 135-214 | 144 (99, 211) | 151 (110, 256) | 135 (82, 175) | 0.020 |
| γ-GT, U/L | 6-42 | 118 (63, 193) | 131 (70, 211) | 100 (57, 160) | 0.032 |
| Albumin, g/L | 35-52 | 32 ± 0.5 | 31 ± 0.6 | 32 ± 0.8 | 0.151 |
| TC, mmol/L | <5.8 | 3.0 ± 0.1 | 3.0 ± 0.1 | 3.1 ± 0.1 | 0.390 |
| TG, mmol/L | <1.7 | 1.3 (0.9, 1.7) | 1.2 (0.8, 1.8) | 1.4 (1.0, 1.6) | 0.563 |
| HDL-C, mmol/L | 1.04-1.55 | 0.51 ± 0.04 | 0.48 ± 0.05 | 0.57 ± 0.07 | 0.309 |
| LDL-C, mmol/L | <3.37 | 2.00 ± 0.13 | 1.90 ± 0.17 | 2.18 ± 0.22 | 0.336 |
| PT, s | 11.5-14.5 | 14.8 (14.0, 16.0) | 15.0 (14.1, 16.2) | 14.3 (14.0, 15.7) | 0.089 |
| APTT, s | 29-42 | 43 ± 0.5 | 43 ± 0.7 | 42 ± 0.8 | 0.266 |
| Fib, g/L | 2-4 | 6.3 ± 0.2 | 6.3 ± 0.2 | 6.4 ± 0.2 | 0.867 |
| D-D dimer, μg/ml FEU | <0.5 | 3.6 (2.1, 5.2) | 3.5 (1.9, 6.1) | 3.9 (2.3, 5.0) | 0.823 |
| Random blood glucose, mmol/L | <11.1 | 7.0 (5.6, 10.8) | 7.0 (5.6, 10.7) | 7.1 (5.5, 11.0) | 0.919 |
| BUN, mmol/L | 2.6-7.5 | 4.5 (3.1, 5.9) | 4.7 (3.1, 6.3) | 4.0 (3.1, 5.6) | 0.307 |
| Creatinine, μmol/L | <1 | 70 (54, 86) | 72 (57, 88) | 69 (53, 77) | 0.143 |
| NT-proBNP, pg/ml | <62.9 | 671.0 (262.5, 2596.0) | 671.0 (238.5, 2393.5) | 1,432.5 (262.3, 3222.5) | 0.684 |
| cTnI , pg/mL | <−34.2 | 3.2 (0.0, 7.4) | 2.7 (0.0, 7.2) | 3.2 (0.0, 7.8) | 0.875 |
| Treatments | | | | | |
| Antibiotics alone | – | 42 (27.3) | 24 (26.4) | 18 (28.6) | 0.763 |
| Antibiotics plus percutaneous drainage | – | 96 (62.3) | 57 (62.6) | 39 (61.9) | 0.926 |
| Antibiotics plus surgical | – | 9 (5.8) | 6 (6.6) | 3 (4.8) | 0.445 |
| Albumin infusion | – | 71 (44.3) | 47 (60.6) | 24 (40.1) | 0.097 |
| Glucocorticoids | – | 47 (30.5) | 33 (36.3) | 14 (22.2) | 0.045 |
| Antiviral drug | – | 11 (7.1) | 6 (6.6) | 5 (7.9) | 0.916 |

**Notes.**

Data are presented as mean ± SE or median (interquartile range) for continuous variables. *P*-values comparing sarcopenia group and non-sarcopenia group are from Student's t test or Mann–Whitney U-test, $\chi 2$ test or Fisher's exact test.

Abbreviations: CRP, C-reactive protein; PCT, procalcitonin; ALT, alanine aminotransferase; AST, aspartate aminotransferase; ALP, alkaline phosphatase; γ-GT, gamma glutamyl transpeptidase; TC, total cholesterol; TG, Triglyceride; HDL-C, high-density lipoprotein cholesterol; LDL-C, low-density lipoprotein cholesterol; PT, prothrombin time; APTT, activated partial thromboplastin time; Fib, Fibrinogen; BUN, blood urea nitrogen; NT-proBNP, n-terminal pro-brain natriuretic peptide; cTnI, cardiac troponin I.

Liu et al. (2023), *PeerJ*, DOI 10.7717/peerj.16055

**Table 3   CT parameters between sarcopenia and non-sarcopenia groups in patients with pyogenic liver abscess.**

| CT parameters | Male | | P | Female | | P |
|---|---|---|---|---|---|---|
| | Sarcopenia | Non-sarcopenia | | Sarcopenia | Non-sarcopenia | |
| SMA (cm$^2$) | 128.28 ± 2.36 | 164.10 ± 2.64 | **<0.001** | 90.16 ± 2.85 | 110.95 ± 2.83 | **<0.001** |
| SMI(cm$^2$/m$^2$) | 43.79 ± 0.75 | 56.27 ± 0.46 | **<0.001** | 35.67 ± 0.46 | 44.73 ± 0.85 | **<0.001** |
| SMD(HU) | 40.84 (34.06, 45.10) | 41.83 (36.94, 47.92) | 0.161 | 36.01 (26.27, 43.32) | 39.74 (28.87, 42.68) | 0.696 |
| IMAT(cm$^2$) | 7.10 (3.29, 10.72) | 6.56 (4.84, 9.16) | 0.877 | 6.49 (2.97, 10.91) | 8.77 (5.51, 13.26) | 0.328 |
| VFA(cm$^2$) | 100.23 (58.81, 159.45) | 144.40 (95.45, 211.70) | **0.022** | 70.72 (45.69, 89.95) | 87.62 (61.53, 136.95) | 0.149 |
| SFA(cm$^2$) | 89.56 ± 5.71 | 117.94 ± 7.72 | **0.004** | 147.29 ± 19.19 | 150.76 ± 14.50 | 0.884 |
| MFI | 1.23 (0.84, 1.56) | 1.21 (0.84, 1.51) | 0.838 | 0.46 (0.42,0.58) | 0.71 (0.44, 0.85) | 0.123 |
| VAI(cm$^2$/m$^2$) | 34.14 (19.23, 54.91) | 49.97 (32.00, 73.52) | **0.027** | 29.13 (19.27, 33.31) | 37.00 (23.18, 55.17) | 0.074 |
| SAI(cm$^2$/m$^2$) | 30.60 (20.97, 42.79) | 40.83 (28.51, 50.59) | **0.003** | 50.73 (40.98, 72.58) | 63.05 (39.13, 75.45) | 0.660 |
| abdominal wall fat thickness(mm) | 13.09 (9.16, 16.58) | 15.48 (11.68, 19.74) | **0.039** | 18.78 (14.69,24.77) | 22.13 (15.53, 27.80) | 0.293 |
| Intra-abdominal fat thickness(mm) | 82.21 (69.81, 98.24) | 86.43 (77.66, 107.58) | 0.147 | 65.39 (43.59, 72.13) | 72.52 (55.70, 85.96) | 0.136 |

**Notes.**

Data are presented as mean ± SE or median (interquartile range) for continuous variables. *P*-values comparing sarcopenia group and non-sarcopenia group are from Student's t test or Mann–Whitney U-test.

Abbreviations: SMA, skeletal muscle area; SMI, skeletal muscle index; IMAT, intramuscular adipose tissue; VFA, visceral fat area; SFA, subcutaneous fat area; MFI, mesenteric fat index; VAI, visceral adipose index; SAI, subcutaneous adipose index; SMD, skeletal muscle density.

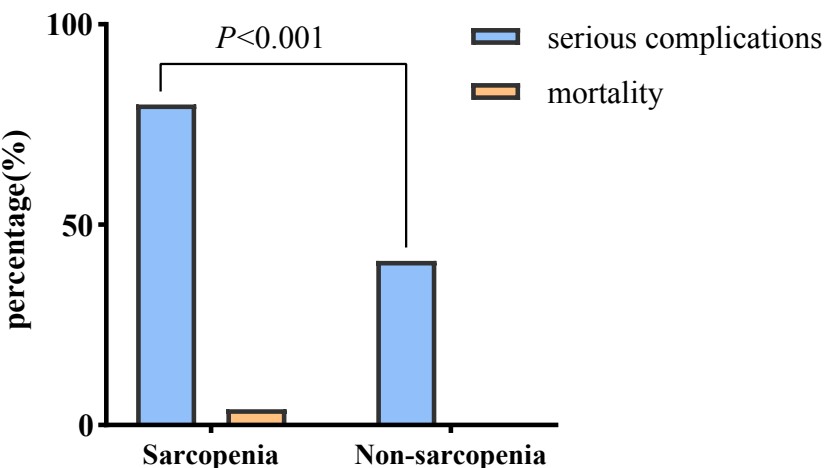

**Figure 2** Comparison of adverse outcomes in the sarcopenia and non-sarcopenia group.

## Independent association between sarcopenia/SMI and adverse outcomes in patients with PLA

Notably, the sarcopenia group of patients with PLA exhibited a higher likelihood of developing serious complications (80.2% *vs.* 41.3%, $P < 0.001$) and experiencing death (4.4% *vs.* 0.0%) than those in the non-sarcopenia group (Fig. 2).

After conducting univariate logistic analysis, we observed that older age ($P = 0.030$), elevated liver function-related indicators such as ALT ($P < 0.001$), AST ($P < 0.001$), PT ($P = 0.010$), and TBIL ($P = 0.002$), elevated inflammatory markers such as CRP ($P = 0.001$) and PCT ($P = 0.032$), elevated creatinine ($P < 0.001$) and BUN ($P = 0.002$), decreased albumin ($P = 0.001$), TC ($P = 0.002$), SMI ($P = 0.006$), and SMD ($P = 0.018$), and the presence of pleural effusion ($P = 0.001$) were correlated with adverse outcomes (serious complications or mortality) in the total subjects, albeit with slight differences across gender subgroups (Table S1). Importantly, SMI showed significantly independent correlation with the adverse outcomes in patients with PLA (odds ratio (OR) 0.787, 95% CI [0.649–0.954], $P = 0.015$) after adjusting for confounding factors (age, gender, ALT, total bilirubin; PT, albumin, CRP, creatinine, pleural effusion) in multivariable analysis (Table 4). In addition, as indicated in Table S2, the multivariate regression analysis conducted in the subgroup of patients with microbial culture results also demonstrated that SMI was independently associated with adverse outcomes of PLA (OR 0.839, 95% CI [0.739–0.951], $P = 0.006$), even after adjusting for confounding factors, including *Klebsiella pneumoniae.*

We further assessed the effect of the sarcopenia in the sub-groups according to gender. As seen in Table 5, in model 3, the ORs were 32.566 (95% CI [1.643–645.343], $P = 0.022$) and 28.327 [1.573–522.290], $P = 0.025$) for male and female subjects, respectively. In addition, as a continuous variable, for every 1 unit increase in SMI, the adjusted ORs of adverse outcomes were 0.864 (95% CI [0.752–0.993], $P = 0.039$) and 0.737 (95% CI [0.546–0.994], $P = 0.046$) for male and female subjects, respectively.

**Table 4** Univariate and multivariate logistic regression analysis for risk factors associated with adverse outcomes in patients with pyogenic liver abscess.

| Variables | Univariate OR (95% CI) | *P* value | Multivariate OR (95% CI) | *P* value |
|---|---|---|---|---|
| Age | 1.029 (1.003, 1.055) | 0.030 | 0.949 (0.836, 1.077) | 0.417 |
| Male/Female | 1.719 (0.834, 3.542) | 0.142 | 4.044 (0.128, 128.029) | 0.428 |
| ALT, U/L | 1.092 (1.056, 1.129) | <0.001 | 1.108 (1.025, 1.197) | 0.010 |
| Total bilirubin, µmol/L | 1.099 (1.036, 1.166) | 0.002 | 0.990 (0.864, 1.134) | 0.887 |
| PT, s | 1.399 (1.085, 1.803) | 0.010 | 1.606 (0.699, 3.690) | 0.265 |
| Albumin, g/L | 0.901 (0.847, 0.960) | 0.001 | 0.864 (0.674, 1.107) | 0.247 |
| CRP, mg/L | 1.010 (1.004, 1.017) | 0.001 | 0.999 (0.987, 1.010) | 0.811 |
| Creatinine, µmol/L | 1.042 (1.021, 1.063) | <0.001 | 1.068 (0.993, 1.148) | 0.077 |
| Pleural effusion | 3.137 (1.572, 6.256) | 0.001 | 4.373 (0.417, 45.838) | 0.218 |
| SMI | 6.195 (2.992, 12.824) | <0.001 | 0.787 (0.649, 0.954) | 0.015 |

Notes.
The covariates included in the multivariable logistic regression analysis were age, gender, ALT, total bilirubin, PT, albumin, CRP, creatinine, pleural effusion, SMI.
Abbreviations:: ALT, alanine aminotransferase; PT, prothrombin time; CRP, C-reactive protein; SMI, skeletal muscle index.

**Table 5** Subgroup analysis based on gender of association of sarcopenia/ SMI with adverse outcomes in patients with pyogenic liver abscess.

| | Model 1 | *P* | Model 2 | *P* | Model 3 | *P* |
|---|---|---|---|---|---|---|
| | OR (95% CI) | | OR (95% CI) | | OR (95% CI) | |
| **Male** | | | | | | |
| Sarcopenia | 4.833 (2.048, 11.404) | <0.001 | 4.721 (1.959, 11.377) | 0.001 | 32.566 (1.643, 645.343) | 0.022 |
| Per 1 unit increase | 0.913 (0.859, 0.971) | 0.004 | 0.914 (0.861, 0.971) | 0.004 | 0.864 (0.752, 0.993) | 0.039 |
| **Female** | | | | | | |
| Sarcopenia | 10.667 (2.387, 47.659) | 0.002 | 10.562 (2.355, 47.375) | 0.002 | 28.327 (1.537, 522.209) | 0.025 |
| Per 1 unit increase | 0.862 (0.760, 0.978) | 0.021 | 0.865 (0.762, 0.981) | 0.024 | 0.737 (0.546, 0.994) | 0.046 |

Notes.
Model 1: Unadjusted; Model 2: adjusted for age; Model3: male subjects adjusted for age, ALT, total bilirubin, albumin, CRP, creatinine, pleural effusion; female subjects adjusted for age, platelet count, ALT, albumin, pleural effusion.
Abbreviations: SMI, skeletal muscle index; ALT, alamine aminotransferas.

## Predictive performance of SMI for adverse outcomes in patients with PLA

ROC analysis was performed to evaluate the relationship between SMI and the prognosis of PLA, and the results are presented in Fig. 3. In male subjects, for the prediction of adverse outcomes in patients with PLA, the AUC value was 0.718 (95% CI [0.617–0.819], $P < 0.001$). The optimal cut-off value of SMI (when the Youden index reaches the maximum) was <52.59 $cm^2/m^2$ with a corresponding sensitivity and specificity of 61.76% and 80.26%, respectively. For female patients with PLA, the AUC value for the prediction of adverse outcomes was 0.714 (95% CI [0.548–0.880], $P = 0.017$, and optimal cut-off was <38.39 $cm^2/m^2$, with a sensitivity of 85.00% and specificity of 65.22%.

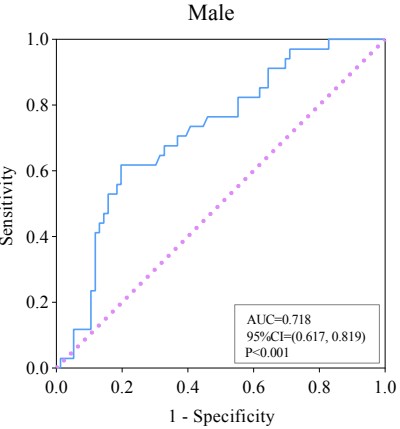
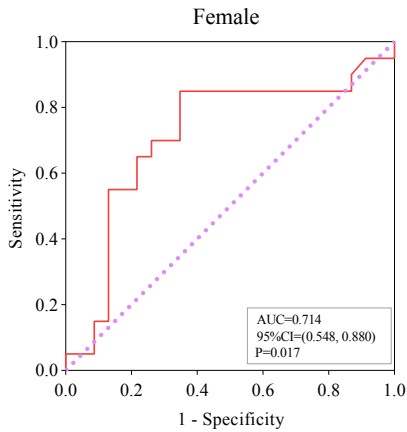

**Figure 3** ROC analysis of SMI for the prediction of adverse outcomes among patients with pyogenic liver abscess.

## DISCUSSION

This study is an attempt to explore the relationship between sarcopenia, as measured by CT, and adverse outcomes in patients with PLA. A total of 154 hospitalized cases with PLA from the same medical center were included retrospectively in this analysis. The sarcopenia group exhibited significantly higher levels of inflammatory markers and liver function-related indicators compared to the non-sarcopenia group. Patients with sarcopenia had a higher risk of serious complications and mortality. In addition, sarcopenia was found to be an independent risk factor significantly associated with adverse outcomes in patients with PLA. ROC analysis also demonstrated that SMI might be a candidate index for predicting the adverse outcomes in PLA patients.

Sarcopenia is a progressive and systemic skeletal muscle disease that carries an elevated risk of adverse outcomes, including falls, physical disability, and mortality. EWGSOP recommends a comprehensive definition of sarcopenia that encompasses low muscle mass in combination with low muscle strength and/or low physical fitness. In assessing muscle mass, CT is considered the gold standard (*Cruz-Jentoft et al., 2019*). Traditionally, sarcopenia has been primarily associated with aging. However, subsequent research findings have indicated that the phenotype of sarcopenia can be caused by various factors beyond aging, suggesting that the development of sarcopenia may initiate early in life (*Dodds et al., 2014*; *Sayer et al., 2008*). In our study, we observed no significant difference in age between the sarcopenia group and the non-sarcopenia group. While the univariate analysis showed a significant association between older age and adverse outcomes, this relationship did not reach statistical significance in the multivariate analysis. This finding aligns with a previous study (*Chan et al., 2022*), which also reported no correlation between age and adverse outcomes in patients with PLA.

Previous studies have confirmed significant correlations between abscess size (*Lee et al., 2021*) and the type of microorganisms (*Chan, Chia, & Shelat, 2022*) with the prognosis of PLA. However, the impact of gas formation on the prognosis of PLA is still uncertain

and subject to debate (*Chan et al., 2020*). In our study, we observed a higher occurrence of *Klebsiella pneumonia* in the non-sarcopenia group. However, neither abscess size, *Klebsiella pneumoniae*, nor gas formation showed significant significance in the univariate regression analysis. Although *Klebsiella pneumoniae* was included in the multivariate regression model, no significant correlation was observed. The discrepancy may be attributed to factors such as our relatively small sample size, differences in the definitions of adverse outcomes, or heterogeneity across different ethnic groups. Despite this, there are also studies that are consistent with our findings, such as Chang Hun Lee's study (*Lee et al., 2021*), which also did not find an independent correlation between *Klebsiella pneumoniae* and adverse outcomes. These results highlight the complexity of the relationship between *Klebsiella pneumoniae* and PLA outcomes. It is crucial to note that the final conclusion remains controversial and further validation with large sample sizes in different populations is warranted. However, whether patients with PLA were complicated with *Klebsiella pneumoniae* infection or not, it was confirmed that sarcopenia was consistently associated with adverse outcomes of PLA after adjusting for multiple variables, including *Klebsiella pneumoniae*. This underscores the clinical significance of SMI as a risk factor in the prognosis of PLA.

Low muscle mass has recently been recognized as part of the definition of malnutrition, reflecting protein malnutrition (*Kang et al., 2018*). Albumin is considered a critical indicator of nutritional status, and it may potentially synergistically contribute to an increased risk of adverse events in association with sarcopenia (*Uemura et al., 2019*). The mechanisms underlying muscle fiber atrophy are intricately linked to the imbalance between muscle protein synthesis and breakdown (*Cruz-Jentoft & Sayer, 2019*). However, our study findings showed no significant difference in albumin levels between the sarcopenic and non-sarcopenic groups. As a meta-analysis reveals, although there is an association between low albumin levels and muscle loss, there is a lack of high-quality evidence in this regard due to variations in the diagnostic methods for muscle loss used in different studies, with the majority of participants being elderly individuals (*Silva-Fhon et al., 2021*). Additionally, albumin levels can be influenced by factors such as infection, liver function, fluid status and other comorbidities, extending beyond just nutritional status (*Zhang et al., 2017*). Furthermore, while albumin is a widely used marker of nutritional status, it may not be the most sensitive or specific indicator for muscle loss. Sarcopenia involves not only inadequate nutrient intake but also complex changes in metabolism and hormonal regulation. In conclusion, there is currently no definitive evidence to consider albumin as a biomarker for muscle loss. In our study, other nutrition-related indicators, such as hemoglobin, lymphocyte count, and creatinine, did not show significant differences between the two groups. The complexity of various factors, such as age, diet, infection, and underlying diseases, may contribute to these outcomes.

It has been determined that malnutrition is very common in patients with various diseases, including PLA, and is closely related to an increased risk of cardiovascular and infection-related mortality (*Sharma et al., 1996*). Nutritional status is linked to immunity, and patients with malnutrition are more prone to infections and complications. (*Keusch, 2003*). Nutritional assessment is, therefore, crucial in patients with PLA. A retrospective
study involving 240 PLA patients showed that lower levels of nutritional risk index were closely associated with poor prognosis of PLA (*Xu, Zhou & Zheng, 2019*).

Sarcopenia is an underestimated but key infection factor (*Schaible & Kaufmann, 2007*), as energy is expended to activate and maintain the immune response during infection. Inflammatory mediators may induce a catabolic state, resulting in increased consumption of arginine from muscle. Some studies have shown that the lack of arginine may impair lymphocyte reaction and limit the availability of complement components (*Beisel, 1996*; *Bronte & Zanovello, 2005*). The relationship between malnutrition and infection is complex. Malnutrition leads to decreased resistance to infection, while infection can exacerbate malnutrition, potentially creating a vicious circle (*Lucidi et al., 2018*). Moreover, muscle is essential for the regulation of energy metabolism such as glucose, heart and respiratory function, and cytokine activity (*Tandon et al., 2017*).

Previous studies have linked the presence of sarcopenia to increased susceptibility to infection, higher risk of complications such as postoperative infection and mortality in patients undergoing surgery (*Kaido et al., 2013*; *Tosteson et al., 2007*), and poor prognosis in patients with sepsis or respiratory failure (*Hung et al., 2021*). In a study of 149 elderly ICU patients with injuries, sarcopenia was an independent predictor of mortality (*Moisey et al., 2013*). Another study of 102 surgical ICU patients by *Mueller et al. (2016)* found that sarcopenia predicted a tendency for longer hospital stays. In conclusion, sarcopenia is closely linked to the prognosis of infection-related diseases.

Current research suggests that changes in body composition in patients with inflammation-related diseases might not be detected by traditional nutritional assessments, leading to the possibility of missing some high-risk patients (*Xu, Zhou & Zheng, 2019*). Additionally, abdominal CT imaging is one of the most important diagnostic tools for patients with PLA and is a necessary examination for almost all patients suspected of having PLA, with most scans include the L3 level to measuring the SMA/SMI. The results of this study indicate that SMI may be used as an alternative indicator to more accurately evaluate the nutritional status of PLA patients and predict their outcomes. This enables early assessment of muscle mass in PLA patients without additional investigation, avoiding unnecessary increases in medical costs and radiation exposure for patients.

Sarcopenia and visceral obesity are considered multifactorial syndromes with various overlapping causes and feedback mechanisms, and are considered strongly interrelated (*Kalinkovich & Livshits, 2017*; *Kohara, 2014*). Previous research has highlighted the complexity of the role of fat in disease and the need for further investigation (*Bouillanne et al., 2009*; *Bronte & Zanovello, 2005*; *Després et al., 2008*). In this study, there was no significant correlation between adipose-related indexes and the prognosis of PLA patients. Interestingly, male patients with sarcopenia had less fat mass, while this was not observed in female patients, possibly due to the biological effects of the difference in fat distribution. Sarcopenia is defined as low SMI, which is normalized from SMA measured on CT images (*Ubachs et al., 2019*). While SMA measurements are influenced by the fat content of skeletal muscle, the SMD indicates the degree of muscle infiltration in skeletal muscle and its measurement does not require standardization for gender or height. Some studies have shown that SMD has independent predictive value for disease prognosis similar to SMI

(*Lucidi et al., 2018*). However, other studies have reported a lack of correlation between SMD and disease prognosis (*Caan et al., 2018*; *Cortellini et al., 2018*). In this study, in patients with PLA, we also did not find a significant relationship between SMD and prognosis, indicating that the role of SMD in disease prognosis remains controversial and needs further investigation.

As the liver is an important metabolic organ of the human body, liver-related diseases and changes in body composition may be may interact and have a close relationship (*Bémeur & Butterworth, 2014*). From a clinical perspective, the results of this study may suggest the importance of preventing or correcting amyotrophy. Therefore, a multidisciplinary team should provide nutritional counseling to the PLA patients, ensuring adequate levels of essential amino acids, and guiding timely and appropriate muscle exercise. However, the extent of this effect on muscle mass is not yet known, as there are many other causes of sarcopenia besides malnutrition (*Lee et al., 2018*). In the future, further intervention research should be conducted to determine whether nutrition and exercise therapy could prevent the progression of sarcopenia and improve the prognosis of patients with PLA.

This is the first study to investigate the association of body composition indicators, especially sarcopenia, with adverse outcomes in PLA patients, using CT to measure muscle and fat-related indicators. Nonetheless, there are some important facts to be acknowledged. Firstly, this was a retrospective study, and an absolute causal relationship could not be established directly based on the results of this study. Secondly, our subjects were recruited from a single center, which may limit the generalizability of our findings to the entire PLA patient population in the region. Thirdly, the data collected in this study consisted of indicators obtained on hospital admission without dynamic observation. Fourthly, ROC analysis indicated that sarcopenia has some predictive power for poor resolution in patients with PLA, while its predictive ability may be limited when used in isolation. Future models should consider SMI in conjunction with other factors and markers to improve the accuracy and reliability of outcome predictions. Finally, the number of subjects included in the study was still insufficient, which restricted the scope of our analysis. Further studies with larger sample sizes, multicenter recruitment, and involving patients of different ethnic origins are needed to provide more comprehensive insights.

## CONCLUSIONS

Sarcopenia is an independent risk factor for adverse outcomes in patients with PLA. SMI, measured by CT imaging, may serve as a valuable tool in risk assessment and hierarchical management for patients with PLA, enabling early identification of those with potentially poor outcomes and facilitating timely and appropriate intervention. The use of CT measurements to assess body composition represents a novel approach, and in the future, it may be necessary to develop SMI cut-off values for sarcopenia with different diseases in order to better comprehend the clinical significance of low muscle mass.

## ACKNOWLEDGEMENTS

The authors thank the staff at the Department of Endocrinology and Medical record, Tongji Hospital, Tongji Medical College, Huazhong University of Science and Technology, and all the patients who participated in the study. We extend our gratitude to Dr. Wenhua Liu from the Clinical Research Center of Tongji Hospital for her valuable guidance on the statistical analysis methods used in our research.

### Funding

This project was funded by the Sen-Mei China Diabetes Research Fund (Z-2017-26-1902 to Zhelong Liu), the Teaching Research Project of Huazhong University of Science and Technology (2022141 to Li Liu), and the Teaching Research Fund of the Second Clinical College of Huazhong University of Science and Technology (2021058 to Li Liu). The funders had no role in study design, data collection and analysis, decision to publish, or preparation of the manuscript.

### Grant Disclosures

The following grant information was disclosed by the authors:
Sen-Mei China Diabetes Research Fund: Z-2017-26-1902.
Teaching Research Project of Huazhong University of Science and Technology: 2022141.
Teaching Research Fund of the Second Clinical College of Huazhong University of Science and Technology: 2021058.

### Competing Interests

The authors declare there are no competing interests.

### Author Contributions

- Li Liu conceived and designed the experiments, performed the experiments, analyzed the data, prepared figures and/or tables, and approved the final draft.
- Shaohua Liu conceived and designed the experiments, performed the experiments, analyzed the data, prepared figures and/or tables, and approved the final draft.
- Meng Hao analyzed the data, authored or reviewed drafts of the article, and approved the final draft.
- Song Hu analyzed the data, authored or reviewed drafts of the article, and approved the final draft.
- Tian Yu analyzed the data, authored or reviewed drafts of the article, and approved the final draft.
- Yunkai Yang analyzed the data, authored or reviewed drafts of the article, and approved the final draft.
- Zhelong Liu conceived and designed the experiments, authored or reviewed drafts of the article, and approved the final draft.
## Human Ethics

The following information was supplied relating to ethical approvals (i.e., approving body and any reference numbers):

The Ethics Committee of Tongji Hospital of Huazhong University of Science and Technology had approved this study. (Ethical Application Ref: TJ-IRB20230118).

## Data Availability

The raw measurements are available in the File S1. According to the value of SMI, all patients were divided into sarcopenia group and non-sarcopenia group for comparative analysis.

## Supplemental Information

Supplemental information for this article can be found online at http://dx.doi.org/10.7717/peerj.16055#supplemental-information.

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
