# Peer review of "Sarcopenia as an important determinant for adverse outcomes in patients with pyogenic liver abscess"

_PeerJ, doi:10.7717/peerj.16055_

## Round 0.1 · original submission · Major Revisions

Both Reviewers made constructive revisions, and I believe that revising this paper based on these comments and suggestions will greatly enhance the quality of the paper and its potential academic impact. Please make the corresponding revisions and resubmit the paper.

Reviewer 1 ·

Basic reporting

This is an interesting retrospective analysis exploring sarcopenia as a determinant of adverse outcomes in patients with pyogenic liver abscess. The background and aims are clearly articulated and the discussion is balanced and informative. The figures and tables are informative.
The manuscript is very well written, but there are a limited number of typos/grammatical issues that need to be rechecked. Abbreviations should be checked to be defined at first use in both abstract and main text and not more than once in the main text.


For instance:
• In the abstract, line 55, no need to define AUC again.
• In methods of abstract, were included is correct.
• In results of abstract, SMI abbreviation should be defined.
• Line 86: mortality of approximately
• Line 101: fight against
• Line 108: There is no need to use ‘’although’’
• Line 114: outcomes of
• Line 116: participants
• Line 137: system included
• Last word of line 137: patients’
• The sentence in the parenthesis at lines 144-146 should be grammatically improved to be better understood.
• Line 163: representatives
• Line 184: complete the word ‘’bdominal’’
• Line 184: what does it mean ‘’anterior and posterior distances’’?
• Lines 230, 231: start all bacterial names with a capital letter.
• Line 232: replace ‘’As to the treatment’’ with another phrase.
• Line 328: SMI abbreviation has been defined once before.
• VFA P-value in table-3 should be in Bold.

Experimental design

Methods and materials were performed technically standard and explained in sufficient detail. The research question is also meaningful and novel.
In the method section please consider the following suggestions:

• Please indicate the type of cultured specimen, which is used multiple times in the text as ‘’bacterial culture’’. Did you mean the culture of aspirated pus? Moreover, about 1.43% of cultured specimens were Fungus, so consider replacing "bacterial culture" with another word.

• Regarding inclusion criteria (Lines 121-126), It was mentioned that ‘’ a) was essential for diagnosis, and b)-e) were non-essential diagnostic criteria’’, If they were not essential diagnostic criteria, you could simply mention only the essential ones.

• As mentioned firstly, CT scan, MRI and US could be used as imaging examinations (Line 122), But in part 3, abdominal CT scan must be available. So you can only include those with available abdominal Ct scans.

Validity of the findings

no comment.

Additional comments

Please indicated the adjusted variables for multivariate analysis in Table 4.

In the image analysis section, indicate muscles include in the SMA.

regarding Table 5, please explain why some variables, which were indicated as risk factors by univariate analysis (in table S1), were not included in the multivariate-adjusted model? For instance, ALT, AST, and PCT were flagged as probable risk factors associated with adverse outcomes in males with pyogenic abscess, but were not included in the multivariate model.



In the supplementary Excel file (raw data), I have noticed the following points that need the author's attention. It seems there must be some typos. If the following patients were included in the study analysis, please re-check their corresponding data and make sure that these probable typos will not affect the study results.

There were some patients with very low temperatures, about 26 ℃ body temperature.

Some patients had "0" number in the serious complications column; however, they have some serious complications in the next following columns.

All patients with arrhythmia had similar pulse rates and heart rates, while there was another patient with a high difference between heart rate and pulse rate (18 vs 76).

I noticed a patient with a Temperature of 41.2 and heart rate of 88, without a history of heart/respiratory disease. Are the body temperature and pulse rate measured at the same time?

Reviewer 2 ·

Basic reporting

The authors present an interesting study on the impact of sarcopenia on clinical outcomes in PLA. They demonstrated that sarcopenia is an independent risk factor for adverse outcomes in PLA. This manuscript has its merits and novelty. However, i would suggest the authors address some of this questions/concerns and revise their manuscript accordingly.

Experimental design

1. The authors present their data set with strict selection criteria and exclusion of patients with pre-existing conditions which may result in sarcopenia e.g. stroke or peripheral neuropathy. This is commendable
2. I understand that sarcopenia was defined radiologically based on previous studies. However, please also include and discuss about the revised European consensus on the definition of sarcopenia - PMID:30312372
3. How many radiologists were involved in analysis of the CT images? Is it only 1 radiologist or 2 independent radiologists? If by 2 radiologists, how were they standardised?

Validity of the findings

1. While multivariate analysis showed a significant effect of sarcopenia on adverse outcomes, ROC analysis showed only fair correlation. Please correlate this with the conclusion
2. Albumin levels are comparable between sarcopenia and non-sarcopenia group. Are the authors able to explain why? Additionally, there should be discussion on the correlation between nutrition level and sarcopenia as these variables are associated. Are there other markers of nutritional status which are used in this study?
3. The authors showed that older age is associated with adverse outcomes but no significance was noted after multivariate analysis. please consider citing this paper which has demonstrated similarly no correlation between age and mortality outcomes in PLA (PMID: 36474543)
4. While the focus of this paper is on sarcopenia, this paper is centered around PLA. Hence, more details should be given regarding the microbiology and radiological characteristics of PLA. This was only briefly described in the results. Notably, while K.pneumoniae was the most common organism between sarcopenia and non sarcopenia, the incidence is quite different (46.81% and 79.17%), which could have accounted for differences in outcomes. Non-Klebsiella pneumoniae PLA has been associated with worse outcomes (except for metastatic complications) - please consider citing PMID: 36145408, and discuss about this point
5. Similarly, no reports about gas formation and size, which are critical factors in deciding need for percutaneous drainage. gas formation is equivocal in determining outcomes and has been debated on (PMID: 32216699, 29536051)

Additional comments

minor comments
- abstract - lines 47 - this is not a cross sectional study. it is a retrospective study
- introduction - line 105 - CT is the first time it appeared in the main text, please spell in full first
- line 108 - there is "a" after 2019, likely typo
- line 143 - unit cannot be seen clearly
- please ensure that all microbiology names are in italics

---

## Round 0.2 · accepted · Accept

Although one Reviewer who previously suggested minor revisions did not respond to my invitation to review the revised article, another researcher who suggested major revisions has approved this revised article for publication.

In addition, I independently evaluated the revised version and was satisfied with the responses and revisions made by the authors. The concerns of the two Reviewers have been well addressed. With the necessary revisions and improvements, the quality of this paper has been significantly improved. I believe that this revised manuscript is ready to be considered for publication in this journal.

Reviewer 1 ·

Basic reporting

No comment

Experimental design

No comment

Validity of the findings

No comment

Additional comments

The manuscript has been revised according to the indicated recommendations. I have no other remarks.